# The TeraFERMI Electro-Optic Sampling Set-Up for Fluence-Dependent Spectroscopic Measurements

**Nidhi Adhlakha** [1], **Paola Di Pietro** [1], **Federica Piccirilli** [2], **Paolo Cinquegrana** [1], **Simone Di Mitri** [1], **Paolo Sigalotti** [1], **Simone Spampinati** [1], **Marco Veronese** [1], **Stefano Lupi** [2,3] and **Andrea Perucchi** [1,*]

1   Elettra-Sincrotrone Trieste S.C.p.A, S.S.14 km 163.5, Basovizza, 34149 Trieste, Italy; nidhi.adhlakha@elettra.eu (N.A.); paola.dipietro@elettra.eu (P.D.P.); paolo.cinquegrana@elettra.eu (P.C.); simone.dimitri@elettra.eu (S.D.M.); paolo.sigalotti@elettra.eu (P.S.); simone.spampinati@elettra.eu (S.S.); marco.veronese@elettra.eu (M.V.)
2   CNR-IOM, Area Science Park Basovizza, 34146 Trieste, Italy; piccirilli@iom.cnr.it (F.P.); stefano.lupi@roma1.infn.it (S.L.)
3   Dipartimento di Fisica, Università "La Sapienza", P.le A. Moro 2, 00185 Roma, Italy
*   Correspondence: andrea.perucchi@elettra.eu

**Abstract:** TeraFERMI is the THz beamline at the FERMI free-electron-laser facility in Trieste (Italy). It uses superradiant Coherent Transition Radiation emission to produce THz pulses of 10 to 100 µJ intensity over a spectral range which can extend up to 12 THz. TeraFERMI can be used to perform non-linear, fluence-dependent THz spectroscopy and THz-pump/IR-probe measurements. We describe in this paper the optical set-up based on electro-optic-sampling, which is presently in use in our facility and discuss the properties of a representative THz electric field profile measured from our source. The measured electric field profile can be understood as the superimposed emission from two electron bunches of different length, as predicted by electron beam dynamics simulations.

**Keywords:** THz spectroscopy; free-electron-lasers; nonlinear optics

## 1. Introduction

The interest in high power pulsed THz sources has seen a steady increase during the last decade. This is due to the great potential which is offered by THz control of matter [1,2]. THz light can indeed be used to tune and modify material's properties in several ways, from the strong acceleration of free charge carriers, to anharmonic lattice distortion and even to ultrafast magnetic switching since magnetic fields in the Tesla range can be accompanying strong THz pulses.

While synchrotrons have always represented a significant source of high brightness THz light for spectroscopy, their use in non-linear THz studies is limited by their relatively low electric fields (<kV/cm). On the other hand, single pass accelerators can store high charge in sub-ps bunches. For this reason, their use for THz control of matter is becoming widespread [3–9].

TeraFERMI is the THz facility based on the FERMI free-electron-laser (FEL) located in Trieste (Italy). The electron bunches that are used by the FERMI free-electron-laser to produce UV and soft X-ray radiation [10] are further exploited by TeraFERMI [11] in order to produce THz waves. The THz pulses emitted by TeraFERMI are coherent due to the so-called superradiance phenomenon. This occurs when the separation between the radiation-emitting electrons is shorter than the wavelength of the light.



In the first part of this paper we will review the basic characteristics and figures of merit of the TeraFERMI source, while on the second part we will mainly describe the electro-optic-sampling (EOS) based experimental set-up that we are routinely exploiting for nonlinear THz spectroscopies at TeraFERMI. In the third section we discuss a typical electro-optic sampling profile measured at TeraFERMI. Interestingly, the measured electric field profile is compatible with the emission from a 100 fs-long current spike in the bunch head of the ps-long entire electron bunch, as previously predicted by electron beam dynamics simulations [12]. The presence of this current spike is due to Coherent Synchrotron Radiation (CSR) wakefields in the dump line and is the origin of the multi-THz emission at TeraFERMI even in the absence of a sub-ps compression of the total electron bunch.

## 2. The TeraFERMI Beamline

The FERMI FEL operates alternatively with two different FEL lines called FEL1 and FEL2, operating between 20–100 nm and 4–20 nm respectively. After the FEL process, both electron lines are merging into one unique beamline before the electron dump. The TeraFERMI source intercepts the electrons at this position, thus allowing operation simultaneously with both FERMI FELs. The THz photons emitted by a Coherent Transition Radiation (CTR) source (see Section 2.1) are propagated along the THz beamline into the FERMI experimental hall, where the THz light is delivered to a dedicated laboratory.

The THz pulses from TeraFERMI are then used to induce non-linear changes in matter. To probe the effects of the THz pulse we use either the same THz photons emitted by TeraFERMI or another probe synchronized to the FERMI master-clock. At present the laboratory is equipped with a femtosecond infrared fiber laser (MENLO C-Fiber 780), which can be operated either on the fundamental at 1560 nm or on its second harmonic at 780 nm. As we will see in the following, the laser can also be used to perform electro-optic sampling measurements thereby probing the time-resolved evolution of the THz electric fields, as well as their spectral content.

### 2.1. Source

Coherent Transition Radiation is the phenomenon occurring when a relativistic electron crosses the boundary between two media of different refractive index. The emitted THz light is particularly bright and displays cylindrical symmetry and radial polarization. Its energy distribution as a function of frequency ($\omega$) and emission angle ($\theta$) can be described by the Ginzburg-Frank formula [13]:

$$\frac{d^2U}{d\omega d\Omega} = \frac{e^2}{4\pi^3\epsilon_0 c}\frac{\beta^2 sin^2\theta}{(1-\beta^2 cos^2\theta)^2},$$ 

(1)

where $\beta$ is the relativistic factor $v/c$. To get an estimate of the emitted intensity one should multiply the Ginzburg-Frank equation (or its generalized version for the near-field case [14]) by the so-called coherence enhancement factor:

$$N[1+N|\int_{-\infty}^{+\infty}\rho(t)exp(-i\omega t)dt|^2].$$ 

(2)

The TeraFERMI source consists in a 1μm-thick Al membrane of 38 mm diameter, oriented at 45° with respect to the electron beam. A wedged diamond window with 20 mm clear aperture is located 80 mm far from the source. The Ginzburg Frank equation allows estimating typical TeraFERMI emission energies ranging from 30 to 100 μJ per pulse, in good agreement with experimental findings [12]. A pictorial view of the TeraFERMI CTR extraction scheme is illustrated in Figure 1.

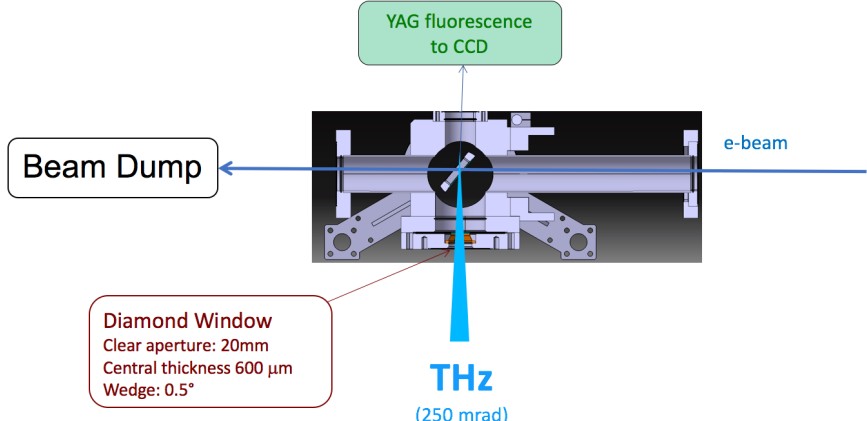

**Figure 1.** Schematics of the TeraFERMI Coherent Transition Radiation (CTR) extraction. The Al membrane intercepts the electron beam at a 45° angle, shortly before the electron beam dump. A diamond window separates the ultra-high vacuum of the electron chamber with respect to the low-vacuum of the TeraFERMI beamline transport system (see Section 2.2). A fluorescent Yttrium Aluminium Garnet (YAG) screen can also be inserted instead of the CTR source (forming a 90° angle with respect to the Al CTR screen). This allows to image the electron beam profile through an additional viewport located opposite with respect to the diamond window.

Under the present ($\sim$1 GeV) energy conditions, the intensity emitted by CTR is comparable to the one that would be obtained through CSR (see Figure 2 in Reference [11]). The main difference between the two types of emission resides in the polarization properties (linear for CSR, radial for CTR) and in the spatial profile. CSR originates from an extended source in the horizontal direction, which may give rise to optical aberrations, unless the vertical and horizontal source emissions are independently focused, by an optimized set of conical and cylindrical mirrors [15]. On the other hand the cylindrical symmetry of CTR radiation can be more easily transported with a simple set of toroidal mirrors (see next Section).

*2.2. Transport System*

The THz beam emitted at the TeraFERMI source needs to be transported along a distance of about 30 m, to reach the dedicated THz laboratory located in the FERMI experimental hall. In order to transport a THz beam over such a considerable length, the beam can not be propagated as a collimated beam but needs to be continuously refocused by a set of 6 toroidal mirrors [16]. The entire beamline, which is separated from the source by an initial diamond window (see Section 2.1), is kept under low vacuum conditions in order to avoid water vapour absorptions. The final window of the beamline is interchangeable so that one can choose the more suitable material (z-cut quartz, sapphire, TPX, etc.) according to the requirements of the experiment. Above 0.3 THz the losses along transport can be totally ascribed to the transmission of the two optical windows. When the TPX window is mounted, the overall beamline transmission is of about 55–60%, which roughly corresponds to the product of the 70% transmission of the initial diamond window and that of the final TPX window (90–95%).

Once reaching the optical table the beam is further refocused through a series of parabolic mirrors finally allowing to steer the THz beam at sample position with the best possible focusing properties. The beam size at focus is 800 μm diameter FWHM, as measured by a pyroelectric camera (Pyrocam IIIHR). This focus was characterised in ambient humidity conditions, when the frequency content at the camera is almost completely restricted below 1 THz.

By inserting a microbolometric camera (i2S TZcam) slightly out of focus it is possible to characterize the typical doughnut shape expected from a CTR source. The left/right asymmetry (see Figure 2), which is also expected from the THzTransport [14] simulation [16], is due to the combination of the 45° orientation of the source radiator and of the 90° off-axis arrangement of the various optical components.

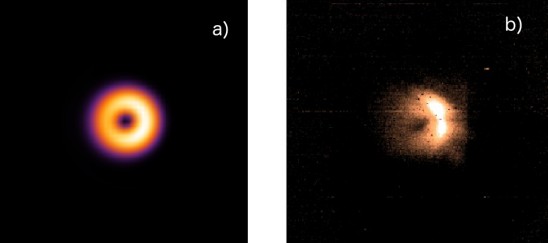

**Figure 2.** (**a**) Simulated beam profile at 1 THz, at focus position. (**b**) Measured beam profile with a microbolometric camera. The typical CTR doughnut shape and the left/right asymmetry is in good agreement with the simulation. In order to achieve the proper spatial resolution needed to visualize the doughnut shape the measurement was performed out of focus, with a larger beam, so that the present comparison is only qualitative. The measurement in (**b**) can not be used to estimate the real beam dimension which was previously measured with the Pyrocam IIIHR THz camera (see text).

*2.3. Beamline Performance*

The overall TeraFERMI performances, were are already summarized in Reference [17]. During standard user operation conditions the beamline produces THz pulses whose energy ranges from 15 to 60 µJ per pulse, with a bunch charge of ∼700 pC and a bunch length of about 1 ps, as measured at the end of the LINAC. This results in energies at sample up to 35 µJ per pulse. The optical spectrum, as measured by a Michelson interferometer, extends up to 4 THz. However, upon optimization of the electron beam compression specific for TeraFERMI, energies at source up to 100 µJ and a spectral extent up to 12 THz were measured [17]. Two characteristic spectra illustrating the performances under standard and THz beamline-dedicated machine conditions are shown in Figure 3.

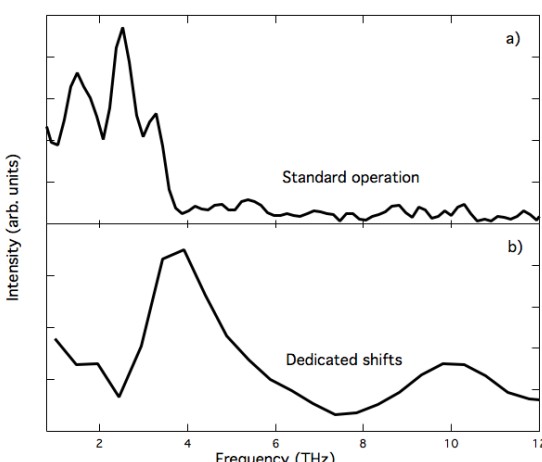

**Figure 3.** Characteristic TeraFERMI spectra, as measured with a Michelson interferometer in (**a**) standard operation conditions and (**b**) dedicated shifts. Figure adapted from Reference [17].

### 2.4. Optical Scheme

The full optical scheme for transmission spectroscopy experiments is depicted in Figure 4. The THz pulse is first polarized and attenuated by a series of three polyethylene grating polarizers. The first and third polarizers select the polarization required for the experiment under study, while the role of the second polarizer is to act as an attenuator, to allow studying the material's fluence dependence.

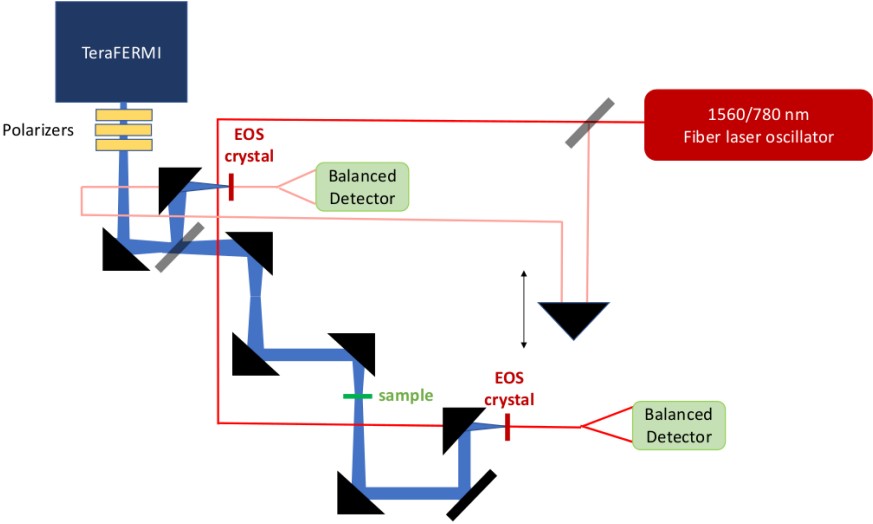

**Figure 4.** Schematics of the THz and optical beam paths. The three initial polarizers are used to set the polarization and attenuate the beam according to experimental requirements. The beam undergoes a series of expansions and recollimations in order to decrease its final size at sample position. Both THz and laser paths are splitted in two to allow for the simultaneous measurement of a reference spectrum.

After being transmitted by the sample the beam is first re-collimated and then focused by a slotted parabolic mirror on the electro-optic sampling crystal. At this point the THz pulse interacts with an optical pulse coming from an infrared fs laser synchronized to the free-electron-laser, as discussed in the next Section.

The main concept underlying EOS is that the THz electric field of the pulse under investigation induces a birefringence in the EOS crystal through the Pockels effect [18]. Thus, if the incoming laser pulse displays circular polarization, which can easily be achieved through a $\lambda/4$ waveplate, after the interaction with a crystal (in our case, a 1 mm thick ZnTe crystal) in presence of THz fields, the polarization will be turned into elliptical. This polarization change can be probed with a Wollaston prism (spatially separating the two orthogonal polarization components) and a balanced detector. Under appropriate approximations, the measured difference signal will then be proportional to the incoming THz field.

In order to allow measuring an online reference of the THz emission spectrum for the source, it is possible to split the THz beam as shown in Figure 4, with the help of a pellicle beam splitter. The infrared laser beam is also being splitted in two different components, one of which is delayed by a suitable amount in order to perform EOS on the reference channel.

### 3. The TeraFERMI Electro-Optic Sampling Set-Up

*3.1. Laser Synchronization*

In order to perform electro-optic sampling measurements (but also to provide time-resolved THz pump-IR probe information), TeraFERMI has been equipped with a commercial mode-locked fiber laser with ultrashort pulses (<100 fs FWHM) and two output wavelength, 780 nm and 1560 nm. A tight synchronization is required to perform time resolved experiments with FERMI THz light. To this purpose the highest performance time reference (LINK) is provided to the beamline optical table, for example, a stabilized optical pulsed signal from the Optical Master Oscillator (OMO) of FERMI.

A Phase Locked Loop (PLL) is used to stabilize the laser phase with respect to the reference. Two error signals feed the loop, one coming from an RF unit (TMU-RF) and one coming from a Balanced Optical Cross-Correlator (BOCC). The RF phase error signal is used for a first, rough synchronization thus making it possible to find the optical phase error signal, with a duration in the order of 1 ps (<0.1% of the pulse period). The optical signal is then used to achieve the final synchronization with few fs RMS jitter in the 10 Hz–10 MHz range.

Inside the BOCC setup we have installed a delay line to provide 6 ns delay (the period of the LINK pulses) with steps of 5 fs. This device is inside the synchronization loop because both the wavelengths of the pump probe laser are derived from the same oscillator, so that when the translation stage is moved the loop corrects the movement by acting on the piezoelectric motor inside the cavity. As a result, the phase of the 1560 nm and 780 nm signals are shifted exactly by the same delay imposed by the delay line. This technique is used to avoid perturbations due to moving elements on the optical path of the beam towards the sample.

The optimization of the LINK dispersion allows to obtain a single clean pulse to be used in the Sum Frequency Generation (SFG) process together with the 1550 nm pump-probe laser pulses inside the BOCC. To this purpose we have developed an optical scheme to measure trough cross-correlation the shape of the LINK pulses. This scheme will be used as BOCC for the synchronization of the laser.

*3.2. Fast Pulse Detection System*

Because of the mismatch between the 78.895 MHz repetition rate of the laser and the 50 Hz repetition rate of the FERMI facility, the detection system employed at TeraFERMI should be able to detect separately all the pulses from the infrared laser. To this aim we are employing a balanced photodetector with fast monitor output up to 350 MHz. The reading of the detector is then performed with the help of a 12 bit, 1 GHz digital oscilloscope. The sampling rate of 10 GS/s allows probing the signal from the photodetector with a 100 ps resolution.

At the maximum sampling rate the oscilloscope acquires the detector signal over a time range of 50 ns. This means that, due to the 78.895 MHz rep rate, we can continuously monitor the signal coming from 4 consecutive laser pulses. Obviously, only one out of the 4 pulses which are present on the oscilloscope's display can be affected by THz light. This means that the remaining three can be used for laser diagnostics. Their signal can be averaged and subtracted to the one which is affected by THz light. In this way we can cancel the residual signal due to laser beam fluctuations and non-perfect balancing between the two detector's channels.

## 4. Electro-Optic Sampling Results

*Measured Electric Field and Spectrum*

An example of EOS measurement during standard beamtime operations is shown in Figure 5. This profile was obtained by scanning the infrared laser with respect to the THz beam at 100 fs steps. For each position of the scanning delay line we have averaged over 10 shots. However, even with a lower (down to 1) number of averages the electric field profile can be clearly reconstructed, thus showing that jitter between THz and laser is not exceeding our step size. We will come back to this point in the following paragraph.

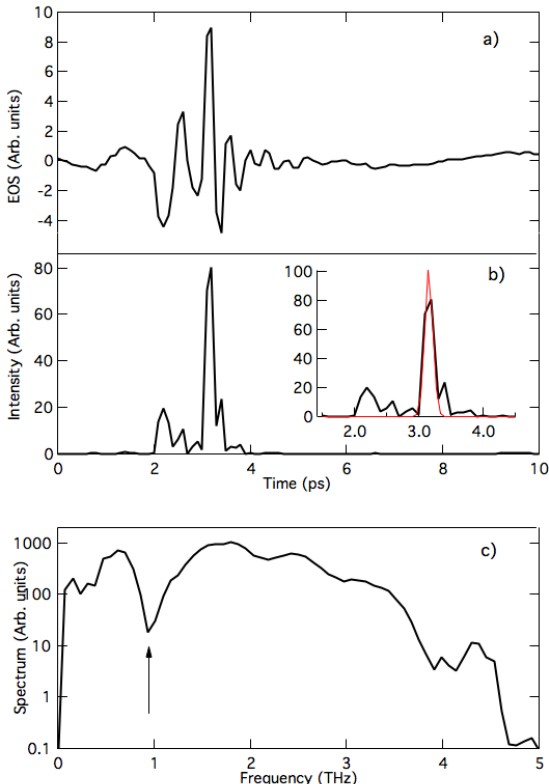

**Figure 5.** (**a**) Representative electro-optic-sampling (EOS) measurement, acquired by averaging 10 shots for each position of the delay line. (**b**) Intensity time profile obtained by evaluating the square of the EOS profile shown in (**a**). The red curve is a Gaussian fit to the main peak. (**c**) Intensity spectrum obtained by calculating the magnitude squared of the Fourier Transform of the spectrum in (**a**).The arrow indicates the dip in the spectrum at about 1 THz which may result from the interference of a long wavelength (∼0.9 THz) and shorter wavelength (∼2 THz) components.

Interestingly, the full energy profile (see Figure 5b) has a duration of about 2 ps, with a main sharp peak containing 60% of the energy of the full pulse. This THz peak is the proof of the presence of a complex electron bunch profile, compatible with the formation of a strong current spike due to the presence of Coherent Synchrotron Radiation (CSR) wakefields, as predicted in Reference [12]. The measured THz peak can be fitted with a gaussian profile, corresponding to a $\Delta t_{FWHM} = 112$ fs FWHM.

The power spectrum related to this electric field shape is reported in Figure 5c, in a logarithmic scale. The spectrum extends up to about 4.5 THz, which roughly corresponds to the maximum detectable signal

in a ZnTe EOS crystal, due to the presence of an optical phonon centered at 5.3 THz [18]. The spectrum shows a pronounced dip at about 1 THz.

The electric field intensity can be determined by employing the standard relationship [19]:

$$E[MV/cm] = 0.39\sqrt{\frac{P[W]}{\pi(r_0[\mu m])^2}}.$$

(3)

The total intensity measured at focus with a calibrated pyroelectric detector in that configuration was $I = 4.8$ µJ. We can then calculate the peak pulse power as $P = 0.58 * I / \Delta t_{FWHM} = 25$ MW. By inserting this value in Equation (3), together with the proper $r_0$ for the measured spatial pulse profile (400 µm FWHM), we end up with an estimate of the THz electric field at 1.65 MV/cm. However it is important to keep in mind that the focus, as measured with the Pyrocam IIHR was characterised in atmospheric humidity conditions, when the presence of spectral components above 1 THz is almost totally absent due to water vapor absorptions. If we assume that the size of the focus scales linearly with the wavelength of the radiation, according to diffraction laws, the spatial profile of the spectral components between 1 and 4 THz may present a radius down to about 100–150 µm FWHM. According to Equation (3), this would imply reaching electric field values as high as 4.5–6 THz. This estimate clearly requires further investigations, for instance by characterising the beam profile under appropriate N2-purging conditions.

The knowledge of the field shape and value is of the highest importance for the interpretation of non-linear phenomena such as saturable absorption or harmonic generation, which depend directly on the electric field, through appropriate scaling laws. Also in pump-probe experiments, the knowledge of the exact shape of the field is important to precisely single out the rise-time and relaxation phenomena occurring as a consequence of the THz excitation.

To achieve a better understanding of the peculiar spectrum and electric field profile of TeraFERMI, we consider two THz single-cycle pulses centered at 0.9 and 2 THz, as depicted in red and blue respectively in Figure 6a. These two pulses mimic the expected composite shape of the bunch profile, being made up of a ∼100 fs high current spike superimposed on a ps-long electron bunch. With a suitable, phenomenological, choice of the relative time delay and phase, the sum of the two pulses yields the electric field profile of Figure 6b, which roughly reminds the EOS result already shown in Figure 5a and reproduced for clarity in Figure 6d. Interestingly the Fourier Transform of this composite pulse displays a remarkable dip at about 1 THz, in agreement with the experimental data, which is due to the interference of the electric fields from the two pulses at distinct frequencies. It is worth emphasizing that the presence of the short current spike is crucial, since it allows extending the TeraFERMI emission in the multi-THz range even when the compression of the electron bunch in the LINAC is not optimized for THz emission, as it is normally the case in parasitic mode operation.

Different machine conditions and in particular different LINAC settings can change the overall bunch compression and the electron energy spread at the entrance of the FEL. This will affect both the duration of the overall electron bunch and the duration and intensity of the current peak. While this can create a different balance in the spectrum between lower and higher frequency components, the presence of a dip at about 1 THz in the TeraFERMI spectra is very common, during normal user operation, thus underlining the validity of the two-component scenario. An EOS characterization of the TeraFERMI emission under optimized conditions would also be of high interest, to provide a better understanding of the spectral structures observed in Figure 3b) with the Michelson interferometer but is unfortunately still missing.

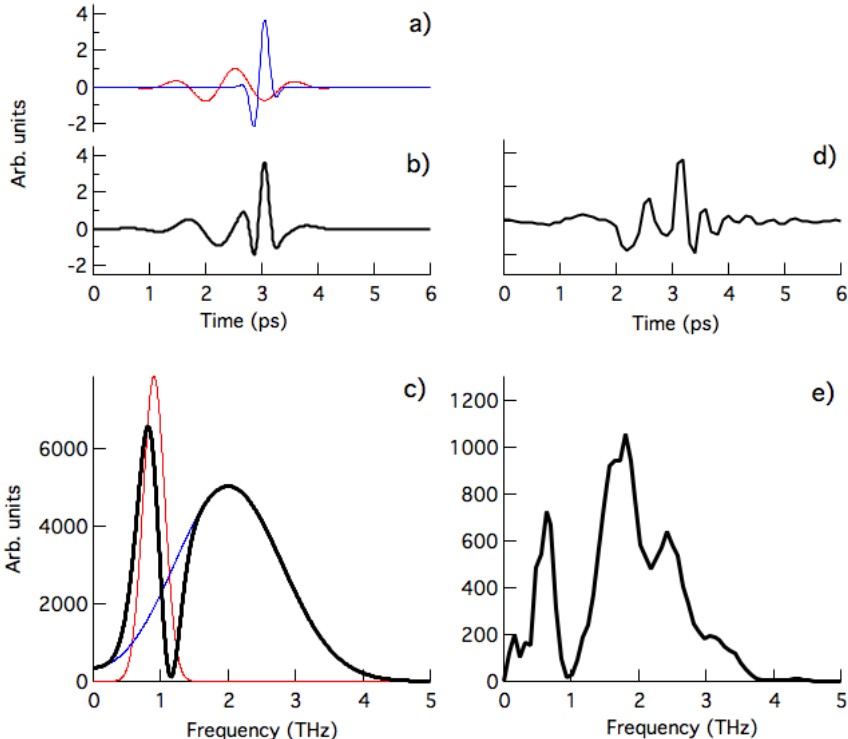

**Figure 6.** (**a**) Simulated THz waveforms centered at 0.9 (**red**) and 2 THz (**blue**), with a Gaussian envelope of width 1 ps and 200 fs respectively. (**b**) Simulated THz waveform resulting from the sum of the two previous contributions (**c**) Intensity spectrum of the 0.9 THz and 2 THz pulses separately and of the sum pulse black. An interference is clearly seen in the black spectra slightly above 1 THz. (**d**) Experimentally measured EOS profile (same as in Figure 5a). (**e**) Experimental spectrum (same as in Figure 5c).

The measured electric field profile as shown above is affected by many broadening effects which are hampering the detection of high frequency components. In particular the laser pulse duration (90 fs RMS) and the jitter in the electron bunch arrival time (<65 fs) are both affecting the measured electric field line-shape. This effect can be mimicked in terms of Gaussian convolutions of the measured electric field profiles. To better quantify the phenomenon we have modelled the electric field shape in terms of discontinuous flat lines, such as their Gaussian convolution provides an electric field profile similar to the one we measured (see Figure 7). The spectrum associated with the modelled electric field provides a much more enhanced signal from 1 to 3 THz. Even though this is obviously an oversimplified model, it may give an idea of the amount of light produced from our source which could still go undetected by our set-up.

Besides the Gaussian broadening effects discussed above other limiting factors could be the velocity mismatch between the THz and optical pulses in the 1 mm thick ZnTe crystal [18] and the already mentioned absorption from ZnTe itself due to the presence of its 5.3 THz optical phonon.

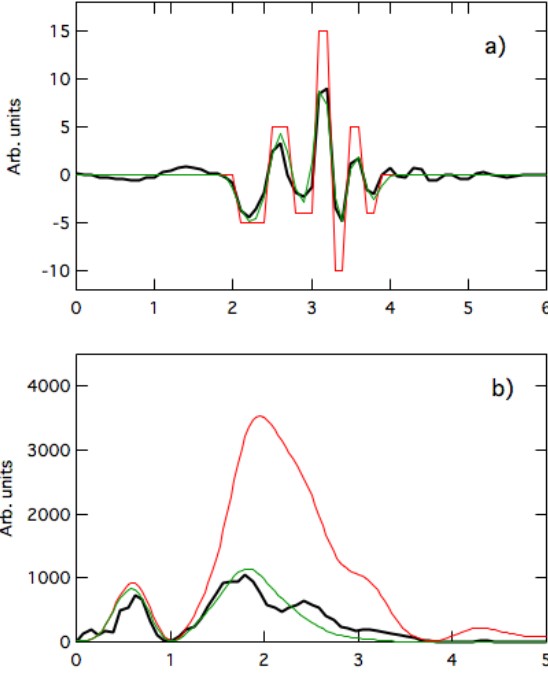

**Figure 7.** (**a**) Experimental EOS profile (black). Model electric field in the form of broken flat lines (red). Convolution of the model field with a 90 fs Gaussian function (green) (**b**) Intensity spectra calculated from the experimental EOS profile (black), from the broken line (red) and from its Gaussian convolution (green). The components at higher frequencies (i.e., from 1 to 4 THz) are clearly suppressed due to the Gaussian convolution which is mimicking the effects of the laser broadening.

## 5. Conclusions

We have reviewed in this paper the properties of the TeraFERMI beamline for THz non-linear and pump-probe studies. In particular we have presented the set-up presently in use for electro-optic-sampling measurements. This allowed characterizing in detail the THz electric field emission from TeraFERMI, which is a crucial information for the interpretation of non-linear THz spectroscopy and pump-probe dynamics. We have found that the usual THz emission from TeraFERMI is the combination of two main components originating by the THz emission from the ps-long electron bunch profile summed to the one from a ∼100 fs long high current spike. This finding is in agreement with previous electron beam dynamics studies [12], pointing to the essential role played by CSR induced wakefields in the high frequency emission properties of TeraFERMI.

**Author Contributions:** Conceptualization, S.L. and A.P.; Investigation, N.A., P.D.P., S.D.M. and A.P.; Methodology, P.C., P.S., S.S. and M.V.; Software, F.P.; Writing—original draft, A.P.; Writing—review and editing, S.L. All authors have read and agreed to the published version of the manuscript.

**Funding:** This research received no external funding.

**Acknowledgments:** We acknowledge F. Vitucci from Crisel Instruments and i2S for providing the TZcam. We are also indebted with F. Novelli for useful discussions and suggestions in the preparation of the optical set-up.

**Conflicts of Interest:** The authors declare no conflict of interest.

## Abbreviations

The following abbreviations are used in this manuscript:

FEL     Free Electron Laser
EOS     Electro Optic Sampling
CSR     Coherent Synchrotron Radiation
CTR     Coherent Transition Radiation
YAG     Yttrium Aluminium Garnet
OMO     Optical Master Oscillator
PLL     Phase Locked Loop
BOCC    Balanced Optical Cross-Correlator
SFG     Sum Frequency Generation

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
