# Peer review of "The TeraFERMI Electro-Optic Sampling Set-Up for Fluence-Dependent Spectroscopic Measurements"

_condensedmatter, doi:10.3390/condmat5010008_

Round 1

Reviewer 1 Report

There are several points to be improved.

Firt of all, the most assertive point of this manuscript must be the analysis of the spectrum from the line 162, but I could not read it in the abstract and introduction sections. The story of this two-component idea must be more emphasised at the begining part of the paper to interest the readers. 

Other query points,

 Similar figure of Fig. 1 is already shown in other papers from your group, and e.g., the layout of CTR at TeraFERMI might be preferable here.  In the line 73, reference 10 is not open access, instead of this manuscript will be open access. The spectrum of TeraFERMI is one of the most interesting information for readers, and that must be more accessible here. In the caption of Fig. 4, averaging shots is 10, not 100 described in line 137. Either will be wrong.   In line 155, why was pyrocam not used as the similar condition of EOS, where above 1 THz is still observed?  I could not find the discussion about Figure 6.   THz is an unit of frequency of electromagnetic wave, so THz wave or FIR light will be better explanations (line 17). 

Author Response

We thank the Referee for his/her competent and insightful report. The manuscript is now corrected according to all of his/her comments and suggestions. In the following we reply to the various points which were raised in the report.

Q: Firt of all, the most assertive point of this manuscript must be the analysis of the spectrum from the line 162, but I could not read it in the abstract and introduction sections. The story of this two-component idea must be more emphasised at the begining part of the paper to interest the readers. 

A: We agree with the Referee, that the interpretation of the spectra in terms of two-components, is indeed the newest and more interesting aspect of the present manuscript. We now mention it both in the abstract and in the introduction (where it was previously completely missing), and we have also expanded its discussion in order to better highlight its important role for TeraFERMI operation.

Q: Similar figure of Fig. 1 is already shown in other papers from your group, and e.g., the layout of CTR at TeraFERMI might be preferable here.  

A: According to the Reviewer's suggestion, we have replaced Figure1 with a schematics of the CTR extraction.

Q: In the line 73, reference 10 is not open access, instead of this manuscript will be open access. The spectrum of TeraFERMI is one of the most interesting information for readers, and that must be more accessible here.

A: In the revised version we added a new Figure, reporting the spectra already discussed in (previous) reference 10, so that the information is now freely available to the reader.

Q: In the caption of Fig. 4, averaging shots is 10, not 100 described in line 137. Either will be wrong. 

A: We thank the Referee for pointing this out. We have now corrected at line 137.

Q: In line 155, why was pyrocam not used as the similar condition of EOS, where above 1 THz is still observed?  

A: When the experiment was perfomed the Pyrocam that we used did not belong to the TeraFERMI beamline, so that unfortunately we were not able to measure the beam profile in all experimental conditions.

Q: I could not find the discussion about Figure 6.   

A: We thank the Referee for pointing this out. (Previous) Figure 6 is now called in the text.

Q: THz is an unit of frequency of electromagnetic wave, so THz wave or FIR light will be better explanations (line 17). 

A: We corrected the manuscript according to the Reviewer's suggestion

A: We thank the Referee for pointing this out. We have now corrected at line 137.

Q: In line 155, why was pyrocam not used as the similar condition of EOS, where above 1 THz is still observed?  

A: When the experiment was perfomed the Pyrocam that we used did not belong to the TeraFERMI beamline, so that unfortunately we were not able to measure the beam profile in all experimental conditions.

Q: I could not find the discussion about Figure 6.   

A: We thank the Referee for pointing this out. (Previous) Figure 6 is now called in the text.

Q: THz is an unit of frequency of electromagnetic wave, so THz wave or FIR light will be better explanations (line 17). 

A: We corrected the manuscript according to the Reviewer's suggestion

Reviewer 2 Report

This paper describes mainly about THz electric field measurement at TeraFERMI facility by EO sampling method.The detail of the THz Beamline is given by references.The EO sampling method itself is already established one.The main contents of this paper seems to be the measurement results of the electric field. Based on the EOS profile, the authors assumed the the bunch contained two components as a simplified model.The data shown seems to be just an example. To conclude this work, how this electric waveform changes with beam operation condition and why the information of the waveform is important to users should be discussed in more detail.   The followings are comments.   Introduction Background, especially references of other accelerator based THz source should be noted.   Line 42 Features of CTR compared with CSR or other types of radiation should be discussed.   Line 46 The emission energy is estimated from 30 to 100uJ per pulse. The assumed bunch charge and bunch length should be given.     Line 54 Above 0.3 the loss… The meaning is not clear. 0.3 means 0.3THz?   Line 64. Ref 9 I could not find the URL of the reference. Reference of a journal paper should be given.   Line 74 Sec.2.4 Since the CTR is radially polarized, the phase of the electric field is flipped in half of the plane. The transverse electric field will be cancelled at the center of the focus. Does this affect the EO sampling system?   Line 84 Material of EOS crystal  (ZnTe) and the thickness should be given in this section.   Line 165 I understand the assumption that the bunch having 100fs spike on ps-long bunch. The blue waveform in Fig 5(a) is asymmetric whereas red waveform is symmetric. Is there any physical assumption for the blue one to be asymmetric?     Line 181 Guassian should be a typo of Gaussian.   Line 186 By this measurement, the characteristics of THz pulse for pump probe experiment is clarified. There should be some discussion how this field shape affects the user experiment.   Line 190 How this bunch shape change depending on FERMI machine condition? Line 72 says that higher energy emitted at optimized operation for TeraFERMI.So, the bunch shape must be affected by machine mode.     Line 193 I  This must be a typing mistake.

Author Response

We thank the Referee for his thorough review, that enabled us to substantially improve the quality of our manuscript. All the Reviewers queries were addressed in the following points.

Q: To conclude this work, how this electric waveform changes with beam operation condition and why the information of the waveform is important to users should be discussed in more detail.

A: We have now emphasized in Section 3.1 the importance of the electric waveform characterization(l.182-186 in the new version). We also discuss in the same section how possible changes in the settings can affect the electric field profile (l.195-207 in the new version).

Q: Introduction Background, especially references of other accelerator based THz source should be noted.  

A: We followed the Referee's suggestion by adding in the introduction a paragraph related to accelerator based THz sources, together with relevant references (Ref 3-9 of the new version)

Q: Line 42 Features of CTR compared with CSR or other types of radiation should be discussed.  

A: A paragraph has been added at the end of Section 1.1 to compare CTR and CSR

Q: Line 46 The emission energy is estimated from 30 to 100uJ per pulse. The assumed bunch charge and bunch length should be given.

A: We now provide this information in the text

Q: Line 54 Above 0.3 the loss… The meaning is not clear. 0.3 means 0.3THz?  

A: We have corrected in "0.3 THz"

Q: Line 64. Ref 9 I could not find the URL of the reference. Reference of a journal paper should be given.

A: The URL is indeed not active anymore. We have replaced the reference with previous Ref.6 (now Ref.14), where information on the THzTransport code can be found

Q: Line 74 Sec.2.4 Since the CTR is radially polarized, the phase of the electric field is flipped in half of the plane. The transverse electric field will be cancelled at the center of the focus. Does this affect the EO sampling system?

A: This is a very interesting point. No major cancellation of the field has been observed while scanning the EOS crystal along the optical axis of THz radiation. On the contrary the EOS peak is maximum at the position where the focus is expected. A full space-time characterisation of the THz field would be extremely useful but is outside of the scope of the present paper.

Q: Line 84 Material of EOS crystal  (ZnTe) and the thickness should be given in this section.  

A: This information is now provided in the 'Optical Scheme' subsection

Q: Line 165 I understand the assumption that the bunch having 100fs spike on ps-long bunch. The blue waveform in Fig 5(a) is asymmetric whereas red waveform is symmetric. Is there any physical assumption for the blue one to be asymmetric?

A: No underlying physical assumption was made here. The phase has been chosen arbitrarily as to better reproduce the experimental data. We have added in the text at l.192, the term 'phenomenological', to better highlight this point.

Q: Line 181 Guassian should be a typo of Gaussian.

A: This has been now corrected

Q: Line 186 By this measurement, the characteristics of THz pulse for pump probe experiment is clarified. There should be some discussion how this field shape affects the user experiment.  

A: This discussion is now present in Sec. 3.1, and briefly recalled in the conclusions.

Q: Line 190 How this bunch shape change depending on FERMI machine condition? Line 72 says that higher energy emitted at optimized operation for TeraFERMI. So, the bunch shape must be affected by machine mode.    

A: The machine conditions, and in particular the LINAC settings can strongly affect the overall bunch compression, and the energy spread at the entrance of the FEL. As a result, these values can change both the duration of the overall electron bunch, and the duration and intensity of the current peak, thus resulting in the significant differences between the spectra in (new) Fig.3 a) and b). Unfortunately no EOS profiles are presently available under these optimized conditions. This discussion is now present in Sec. 3.1

Q: l This must be a typing mistake

A: This is an error in Latex compiling that we were not able to fix. We hope this mistake can be corrected in the editorial process.

Round 2

Reviewer 2 Report

I understand that the authors answered all the comments. The readability of the paper is improved with additional informations. Although some answers are not clear, I understand that they are out of the main scope of this paper. I agree for publication after some minor corrections.       Some minor corrections.   line 33.   a the origin   -> the origin   line 55-60% -> 55-60% for diamond case?   Line 231.  I -> removed

Author Response

We thank the referee for his review. In the following we answer the minor points which were raised. 

Q: line 33.   a the origin   -> the origin  

A: This has been now corrected 

Q: line 55-60% -> 55-60% for diamond case?  

A: This corresponds to an initial diamond window, and a final TPX window. The text was not clear on that point, also because the presence of the diamond window was mentioned in the previous section, but not recalled in the present paragraph. We have slightly rephrased the paragraph in order to make this point clearer.

Q: Line 231.  I -> remove

A: We were not able to remove this character. We hope this mistake can be corrected in the editorial process.